# Inter-species gene flow drives ongoing evolution of *Streptococcus pyogenes* and *Streptococcus dysgalactiae* subsp. *equisimilis*

Ouli Xie[1,2], Jacqueline M. Morris[3], Andrew J. Hayes[3], Rebecca J. Towers[4], Magnus G. Jespersen[3], John A. Lees[5], Nouri L. Ben Zakour[6], Olga Berking[6], Sarah L. Baines[7], Glen P. Carter[7], Gerry Tonkin-Hill[8], Layla Schrieber[9], Liam McIntyre[3], Jake A. Lacey[1], Taylah B. James[3], Kadaba S. Sriprakash[10,11], Scott A. Beatson[6], Tadao Hasegawa[12], Phil Giffard[4], Andrew C. Steer[13], Michael R. Batzloff[10,14], Bernard W. Beall[15], Marcos D. Pinho[16], Mario Ramirez[16], Debra E. Bessen[17], Gordon Dougan[18], Stephen D. Bentley[18], Mark J. Walker[6,19], Bart J. Currie[4], Steven Y. C. Tong[1,20], David J. McMillan[21,22] & Mark R. Davies[3,22] ✉

*Streptococcus dysgalactiae* subsp. *equisimilis* (SDSE) is an emerging cause of human infection with invasive disease incidence and clinical manifestations comparable to the closely related species, *S. pyogenes*. Through systematic genomic analyses of 501 disseminated SDSE strains, we demonstrate extensive overlap between the genomes of SDSE and *S. pyogenes*. More than 75% of core genes are shared between the two species with one third demonstrating evidence of cross-species recombination. Twenty-five percent of mobile genetic element (MGE) clusters and 16 of 55 SDSE MGE insertion regions were shared across species. Assessing potential cross-protection from leading *S. pyogenes* vaccine candidates on SDSE, 12/34 preclinical vaccine antigen genes were shown to be present in >99% of isolates of both species. Relevant to possible vaccine evasion, six vaccine candidate genes demonstrated evidence of inter-species recombination. These findings demonstrate previously unappreciated levels of genomic overlap between these closely related pathogens with implications for streptococcal pathobiology, disease surveillance and prevention.

*Streptococcus dysgalactiae* subspecies *equisimilis* (SDSE), a beta-hemolytic *Streptococcus* normally possessing the Lancefield group C/G antigen, is an emerging cause of human disease with recently reported incidences of invasive disease comparable to or surpassing that of the closely related and historically important pathogen, *Streptococcus pyogenes* (group A *Streptococcus*)[1–8]. SDSE and *S. pyogenes* occupy the same ecological niche and possess overlapping disease manifestations including pharyngitis, skin and soft tissue infections, necrotising fasciitis, streptococcal toxic shock syndrome and osteoarticular infections[9,10].

Gene transfer between SDSE and *S. pyogenes*, including housekeeping multi-locus sequence typing (MLST) loci, major virulence factors including the *emm* gene, and antimicrobial resistance (AMR) determinants has been reported[11–17]. Inter-species transfer of genes encoding the serogroup carbohydrate have led to SDSE isolates which express the group A carbohydrate[15,18]. Transfer of accessory virulence or AMR genes is thought to occur by cross-species exchange of mobile genetic elements (MGEs) such as prophage or integrative conjugative elements (ICE)[19–21]. However, the mechanism that enables the

exchange of genes present in all strains of a species, termed 'core' genes, is not well understood. The extent of genetic exchange between these two pathogens has not been defined within a global population genomic framework.

An analysis of the population structure of a globally diverse collection of *S. pyogenes* genomes provided insights into the drivers of population diversity and global utility of candidate vaccines[22]. In contrast, studies of SDSE whole genome diversity have generally been limited to local jurisdictions[7,23–25]. With increasing efforts to develop a *S. pyogenes* vaccine[26,27], an improved understanding of the overlap and extent of genetic similarity between human isolates of SDSE and *S. pyogenes* in a global context is required. Here, we have compiled a globally diverse collection of 501 SDSE genomes isolated from human hosts. Through a gene synteny-based approach, we conducted a systematic analysis of the population genetics and pangenome of SDSE. Using the same framework, we compare SDSE to a previously published global *S. pyogenes* dataset[22] to reveal extensive genomic overlap between the two closely related pathogens including genes encoding candidate *S. pyogenes* vaccine antigens.

## Results

### A globally diverse SDSE genomic database

To assess SDSE population diversity (Fig. 1), we compiled a genomic database of 501 geographically distributed SDSE isolates from 17 countries including 228 newly reported genomes with one new complete reference quality genome, NS3396[28] (Supplementary Data 1a). All publicly available SDSE genomes on NCBI SRA and RefSeq on 4 May 2022 were included after quality control (see Methods) with manual down-sampling of sequences from studies which selectively focused on limited lineages. The database includes 53 *emm* sequence types, 88 *emm* sub-types, and 129 MLSTs.

### Global SDSE population structure

A phylogeny of the global SDSE population was constructed using a recombination aware pipeline implemented in Verticall. The minimum evolution phylogeny demonstrated a deep radially branching structure forming multiple distinct lineages similar to that observed for the global population structure of *S. pyogenes*[22] (Fig. 2a and Supplementary Fig. 1).

Whole genome clustering of the global SDSE population using PopPUNK[29] detected 59 distinct population clusters (akin to 'lineages' or 'genome clusters') (Fig. 2a and Supplementary Fig. 2). These genome clusters were geographically dispersed and were highly concordant with the inferred phylogeny.

Our data show a limited concordance between the inferred whole genome population structure and the classical SDSE molecular epidemiological markers, *emm* type and MLST (Supplementary Fig. 3). Of the 24 *emm* types represented by three or more isolates, 11 (46%) were present in multiple genome clusters. Of the 27 genome clusters with three or more isolates, 18 (67%) contained two or more *emm* types.

MLST was in stronger agreement with the inferred genome clusters than *emm* type. Nevertheless, the largest genetic distance between isolates in the same MLST was frequently greater than that separating isolates from different MLST types (Supplementary Fig. 3d). Supporting this observation, 67% (18/27) of genome clusters with three or more members contained more than one MLST. While many of these differed at only a single MLST locus, single locus variant MLST clonal complexes delineated distant lineages poorly and were present across multiple distinct genetic backgrounds (Supplementary Fig. 3e). These findings indicate that while *emm* and MLST typing for SDSE have been useful epidemiological markers in jurisdictional or regional contexts, they have limitations when defining global SDSE evolution and diversity. The presented whole genome clustering scheme implemented in PopPUNK provides a solution to the limitations of *emm* and MLST at the global population level[29].

The Lancefield group carbohydrate was predicted by the presence of a 14 gene carbohydrate synthesis gene cluster in group C SDSE[18], 15 gene cluster in group G[15] and 12 gene cluster in group A *Streptococcus*[15] (Supplementary Fig. 4a) in a conserved genomic location immediately upstream of *pepT*. Of the 27 genome clusters which contained three or more isolates, 16 consisted of only group G isolates and 9 consisted of only group C (Supplementary Fig. 4b). Two genome clusters consisted of isolates with more than one group carbohydrate. The Lancefield group A carbohydrate, normally associated with *S. pyogenes*, was found in isolates from 4 different genome clusters (Fig. 2a), including one genome cluster that also contained isolates with the group C and group G antigen. These findings suggest that while closely related isolates generally express the same Lancefield group carbohydrate antigen, horizontal transfer of the group carbohydrate locus can occur, including the group A carbohydrate locus normally associated with *S. pyogenes*.

Bayesian dating of the most recent common ancestor (MRCA) of the two most well sampled genome clusters using BactDating[30] with recombination masked by Gubbins[31], revealed emergence of these lineages within the last 100 years. A genome cluster containing members present in 7 countries and inclusive of *emm* type *stG*6792/ST 17 isolates, until recently the most frequent cause of invasive disease in Japan[7,10,32], had an estimated root date in the mid-1970s (credible

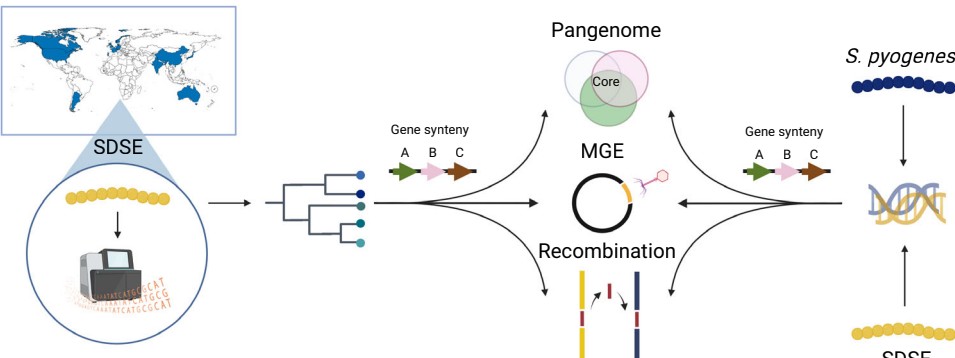

**Fig. 1 | Workflow to characterise the global population structure of *Streptococcus dysgalactiae* subsp. *equisimilis* (SDSE) and its overlap with *S. pyogenes*.** A globally diverse collection of 228 SDSE isolates were whole genome sequenced using Illumina short read sequencing and collated with publicly available genomes to form a database of 501 global sequences. An analysis of the SDSE population structure was undertaken followed by a systematic analysis of the pangenome, evidence of homologous recombination, mobile genetic elements (MGEs) and MGE insertion sites using pangenome gene synteny contextual information. This framework was then compared with a previously published global database of 2,083 *S. pyogenes* genomes by merging pangenomes accounting for shared gene synteny to reveal extensive overlap at the level of shared genes, homologous recombination and MGEs[22]. Figure created with BioRender.com.

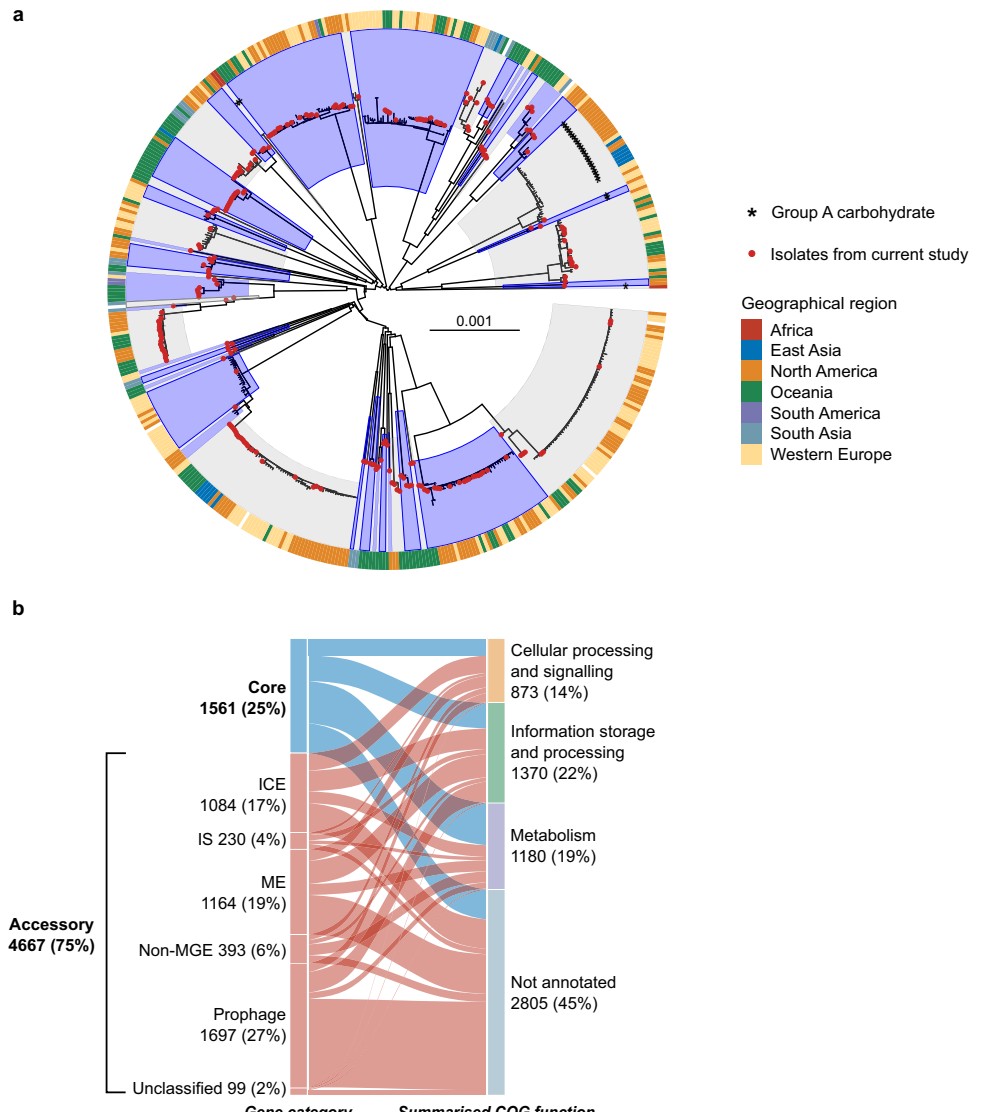

**Fig. 2 | Global population structure and pangenome composition of *Strepto-coccus dysgalactiae* subsp. *equisimilis* (SDSE). a** Minimum evolution phylogenetic tree of 501 SDSE isolates using recombination masked genomic distances. Fifty-nine distinct whole genome clusters are highlighted by alternating blue and grey shades from internal nodes. Newly published sequences from this study are high-lighted by red points at the tips of the tree. Isolates carrying the group A carbo-hydrate are marked by asterisks. The outer ring around the tree denotes geographical region of isolation. **b** Alluvial plot of the SDSE pangenome with categorisation of core (present in ≥99% of isolates) and accessory genes into summarised COG functional groups. Accessory genes are further characterised based on the type of element they are most frequently found. Genes associated with prophage and prophage-like elements were grouped together. Genes asso-ciated with mobile genetic elements (MGE) that could not be classified further were assigned to mobility elements (ME). A small number of unclassified accessory genes were present only adjacent to contig breaks without an integrase and could not confidently be classified into a MGE or non-MGE category. ICE, integrative con-jugative element; IS, insertion sequence; non-MGE, non-mobile genetic element. Source data are provided as a Source Data file.

interval CrI 1960–1991, Supplementary Fig. 5a) suggesting recent emergence and global dispersion. A genome cluster consisting of isolates with *emm* type *stG*62647, an *emm* type described to be increasing in frequency in multiple jurisdictions[24,33], had an estimated root date in the early 1930s (CrI 1855–1969, Supplementary Fig. 5b). These findings suggest that major genome clusters in this global database represent modern, globally disseminated lineages.

Virulence genes or regulators known to influence expression of virulence genes were enriched in a subset of genome clusters. Isolates were screened for the presence of accessory toxins including DNAses, superantigens, adhesins, immune escape factors and regulators described in SDSE or *S. pyogenes* (Supplementary Data 1b). The FCT locus (termed for fibronectin-collagen-T-antigen) is a major adhesion virulence determinant in *S. pyogenes* but has not been experimentally characterised in SDSE. Homologous regions have been found in SDSE

including a second variably present locus containing only putative pilus genes which we have termed here accessory/secondary FCT to prevent confusion with *S. pyogenes* FCT types[34]. We found that certain genome clusters were enriched for accessory factors including: chro-mosomally encoded exotoxin *speG*, immune escape factor *drsG*, adhesin *gfbA* and an accessory/secondary FCT locus, presence of the negative regulatory *sil* locus, and three accessory prophage strepto-dornase genes *spd3*, *sda1* and *sdn* (all *p* < 0.001, $\chi^2$ test of indepen-dence). In particular, the accessory FCT locus[34], *sil* locus, *gfbA*, *speG* and *drsG* were frequently either present or entirely absent from dif-ferent genome clusters. The primary FCT locus was present in almost all isolates (500/501) while the accessory FCT was present in 84% (421/501) of isolates. These findings suggest that the SDSE genetic popu-lation structure is characterised by distinct virulence repertoires. Iso-lates from clinically defined invasive or non-invasive infection were

intermixed in the 10 most frequently sampled whole genome clusters. However, non-random sampling prevented associations with clinical manifestations from being inferred.

### Signatures of recombination in the SDSE core genome

The SDSE pangenome consisted of 6228 genes of which 1561 were core genes and 4667 were accessory genes, of which 3849 genes were present in <15% of genomes (Fig. 2b). While several SDSE genes have previously been reported to be recombinogenic, we next aimed to infer the number of SDSE core genes with signatures of recombination in their evolutionary history using fastGEAR[35]. Of 1543 core genes (excluding 18 genes that had over 25% gaps in their alignment), 837 (54%) genes had a recombinatorial signature (Supplementary Data 2a). These notably included all seven MLST genes. These findings correlate with uncertainties when classifying isolates using MLST compared to whole genome clusters.

To further quantify the contribution of recombination to SDSE population diversity, we measured the ratio of recombination-derived mutation vs vertically inherited mutation (r/m) using Gubbins[31] for the 12 largest genome clusters (385 isolates). The median r/m per genome cluster was 4.46 (range 0.38–7.05) which was comparable to a r/m of 4.95 estimated previously for *S. pyogenes*[22] (Supplementary Table 1). The median recombination segment length was 4647 bp (range 6–97789 bp) and the median number of events per genome cluster was 42 (range 3–64), again similar to that previously described for *S. pyogenes*[22], supporting multiple small fragments of homologous recombination as a major source of genetic diversity in SDSE.

### Contribution of the accessory genome to SDSE diversity

To investigate the contribution of MGE and non-MGE genes to accessory genome diversity, an identification and characterisation workflow was developed using genome synteny and a classification algorithm adapted from proMGE[36]. Segments of accessory genes (hereafter referred to as 'accessory segments') were classified as prophage, phage-like, ICE, insertion sequence/transposon (IS) or non-MGE based on the presence of MGE-specific integrase/recombinase genes and prophage or ICE structural genes (Supplementary Fig. 6a). Accessory segments were classed as mobility elements (ME) when insufficient information was present to classify a segment—such as with degraded elements, complex nested elements or elements fragmented by assembly breaks with insufficient contextual information. Genes were classified into MGE categories based on the frequency of their presence in different elements (Fig. 2b). Prophage elements contributed the largest number of genes to the accessory genome (36%, 1697/4667 genes) followed closely by ME (25%, 1164/4667 genes) and ICE (23%, 1084/4667 genes). Non-MGE accessory genes constituted 8% (393/4667 genes) of the accessory genome. A mean of 0.8 prophage (range 0–5), 1.1 phage-like (range 0–4), 1.1 ICE (range 0–4) and 4.7 ME (range 0–10) were found per genome. MGE counts were similar when restricted to four complete SDSE genomes (mean 1.3 prophage, 1 phage-like, 1 ICE, 2.8 ME). A mean of 10.5 IS/transposon elements (range 3–30) were found per genome but was likely to be an underestimation as parameters used to construct the pangenome resulted in the removal of infrequent genes at genome assembly breaks, which are commonly associated with IS/transposons.

Using genome location defined between two syntenic core genes, MGE chromosomal 'insertion regions' were mapped in the SDSE dataset. A total of 55 insertion regions were found in SDSE (Fig. 3a, Supplementary Data 3a). Prophage or phage-like elements were found at 32 insertion regions (58%), ICE at 21 regions (38%), and 10 regions (18%) were occupied by either prophage or ICE. Twelve regions (22%) were occupied by elements classified as ME only, reflective of the limitations imposed by sequence breaks around MGEs in draft genomes. The MGE insertion regions were broadly occupied across SDSE genome clusters (Supplementary Fig. 6b).

### Conservation of MGE insertion regions across SDSE and *S. pyogenes* are associated with shared elements

Applying the same accessory identification and categorisation workflow to the 2083 *S. pyogenes* genomes published previously[22] enabled a systematic comparison of MGE and associated insertion sites both within and between the two species. An average of 1.9 prophage (range 0–6), 0.4 phage-like (range 0–3), 0.2 ICE (range 0–2), and 1.9 ME (range 0–7) were found per *S. pyogenes* genome. Compared to SDSE, *S. pyogenes* isolates had more prophage elements but less ICE ($p < 0.001$, Wilcoxon rank sum).

Overlaying the SDSE and *S. pyogenes* pangenomes while accounting for genome synteny, 31 previously published[37,38] and 13 new *S. pyogenes* MGE insertion regions (31 prophage, 9 ICE, and 2 mixed prophage and ICE) were mapped and compared to 55 insertion regions in SDSE (Supplementary Data 3c). Of these, 16 (29% of SDSE insertion regions) insertion regions were shared across the species (Fig. 3a). At the 16 cross-species insertion regions, 1443 accessory genes (54% of SDSE and 59% of *S. pyogenes* accessory genes at these regions) were shared across species, suggesting likely shared MGEs at these regions. At insertion regions which were not conserved across the two species, 816 accessory genes (34% of SDSE and 42% of *S. pyogenes* accessory genes) were shared, significantly less than the proportion at conserved insertion regions ($p < 0.001$, $\chi^2$ test).

Hypothesising that shared MGE insertion regions may provide a basis for shared prophage and ICE between species, all completely assembled and fragmented putative MGEs > 10–15 kb at the 16 cross-species insertion regions were examined. A total of 3335 fully assembled and 295 high-quality fragmented putative MGEs, 710 from SDSE and 2920 from *S. pyogenes*, were extracted. To identify similar elements within this database, we used mge-cluster[39] to cluster ICE, ME and prophage elements which overcomes limitations of sequence homology-based approaches which are restricted by the modular and highly recombinogenic nature of MGEs. Using this approach, 40 clusters containing 2897 MGEs were identified of which 10 (25%) ICE and prophage clusters across 13 insertion regions were found in both species (Fig. 3b). The clusters were generally well defined and separated by MGE type (Supplementary Fig. 7). Of the 40 clusters, 14 were present at more than one insertion region (median 1, range 1–7) indicating broad sharing of MGE clusters across these common insertion regions and between species.

We further examined MGE clusters for examples of shared near-identical elements (at least >89% nucleotide identity and coverage). Four MGEs (prophage, ICE and nested IS/transposon elements) carrying streptodornase genes *sda1* and *spd1*, the exotoxin gene *speC*, and AMR genes *ermB*, *ant(6)-Ia* and *aph(3′)-IIIa* were detected across both species (Table 1, Supplementary Fig. 8). A completely assembled prophage element that was near-identical (>99% nucleotide identity) to the *S. pyogenes* M1T1 prophage φ5005.3 was shared across 12 genomes in SDSE and 104 genomes in *S. pyogenes* (Fig. 3c). The M1 prophage φ5005.3 carries the streptodornase gene *sda1*, an extracellular virulence factor thought to have been acquired during the emergence of the global M1T1 *S. pyogenes* lineage[40,41]. To investigate the distribution of this prophage further in SDSE, including MGEs fragmented by contig breaks, co-presence of the same integrase and *sda1* allele was examined across the SDSE database. An additional 8 genomes with co-carriage of these genes at the same insertion region were found. The φ5005.3 prophage was inferred to be present in SDSE isolates spanning 10 different genome clusters across seven countries (Australia, Canada, China, Denmark, Norway, Portugal and USA), over a 20 year period (1999 to 2018). Within these genome clusters, a mean of 28.8% of isolates carried the φ5005.3 phage (range 1.5–100%). These findings indicate that MGEs carrying virulence and/or AMR determinants can cross species boundaries including evidence of dispersal into more than one global SDSE lineage.

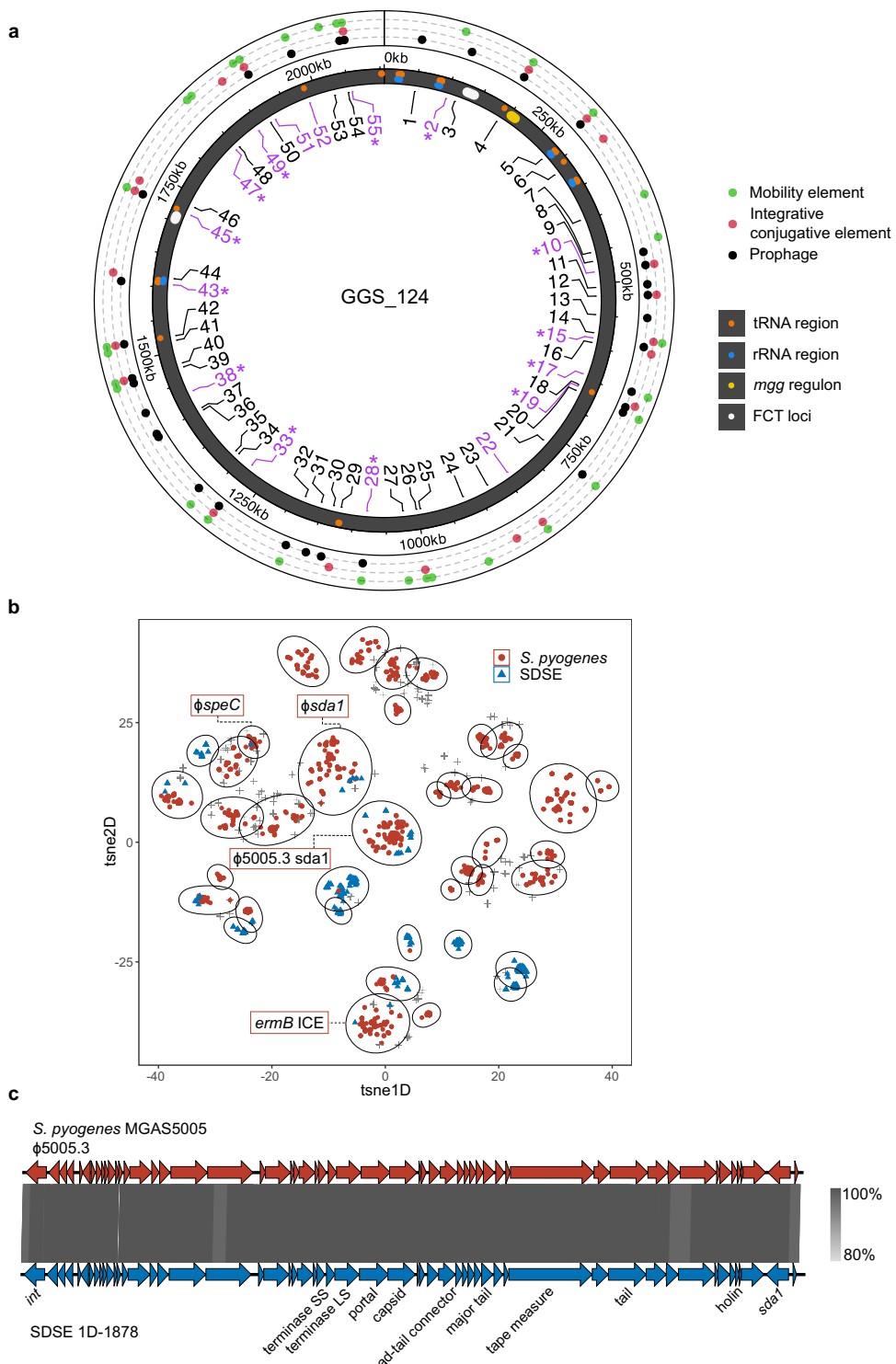

Despite the extensive overlap in accessory genes and MGEs between SDSE and *S. pyogenes*, notable differences in virulence genes were found. The gene encoding the *S. pyogenes* cysteine proteinase SpeB, a core virulence factor, and its regulator *ropB* were absent in SDSE. Additionally, genes encoding the serine proteinase SpyCEP, intact hyaluronic acid capsule synthesis operon *hasABC*, superantigens SpeA, SpeH, SpeI, SpeJ, SpeK, SpeL, SpeM, SpeQ, SpeR, Ssa, Smez, and the gene encoding phospholipase A2 (Sla) were absent in all isolates in this dataset (Supplementary Data 1b). The difference in virulence repertoire of the two species may contribute to phenotypic and virulence differences between the two organisms.

### Extensive overlap and exchange in the core genome between SDSE and *S. pyogenes*

To examine core genome overlap across SDSE and *S. pyogenes*, we found that 1166 core genes comprising 75% (1166/1547) of the SDSE core genome and 88% (1166/1320) of the *S. pyogenes* core genome were shared in the merged pangenome (Fig. 4a, Supplementary Data 2b). A small number of genes/coding sequences were combined when merging pangenomes, resulting in a slightly smaller merged core pangenome compared to SDSE or *S. pyogenes* alone.

To investigate cross species recombination in these core genes, 1166 genes that were core in both species and with <25% length

**Fig. 3 | Clustering and genome localisation of *Streptococcus dysgalactiae* subsp. *equisimilis* (SDSE) and *S. pyogenes* mobile genetic elements (MGEs). a** Location of SDSE MGE insertion regions relative to the GGS_124 reference genome (NC_012891.1). Insertion regions are labelled from 1–55 of which 16 (29%) were shared with *S. pyogenes* (highlighted in purple). Shared insertion regions at which MGE clusters were also shared across species are highlighted with asterisks. The outer ring indicates the type of element detected at each insertion region in SDSE. tRNA and rRNA regions, the *mgg* regulon containing the *emm* gene, and the two FCT loci in GGS_124 are marked on the genome representation. **b** Clustering of 3,630 MGEs at 16 shared insertion regions across SDSE and *S. pyogenes*. Each cluster is outlined by an ellipsoid. Of the 40 MGE clusters, 10 (25%) were shared across species. Clusters which contained notable examples of near-identical (>89% nucleotide identity and coverage) cross-species MGEs are labelled: the global M1T1 φ5005.3 prophage carrying the Sda1 streptodornase (104 *S. pyogenes*, 12 SDSE),

φ*sda1* refers to a prophage carrying a *sda1* allele 95% similar to φ5005.3 (2 *S. pyogenes*, 1 SDSE), φ*speC* refers to a previously described prophage carrying *speC* and *spd1*[18] (2 *S. pyogenes*, 1 SDSE), *ermB* ICE refers to a complex nested ICE and IS/ transposon element carrying multiple AMR genes including *ermB* (1 *S. pyogenes*, 1 SDSE). These MGEs represent a subset of elements within each respective cluster. **c** Genome architecture and comparison of the *S. pyogenes* M1T1 prophage φ5005.3, with a near identical prophage found in group G SDSE isolate 1D-1878 at insertion site 45. 1D-1878 was isolated from a case of invasive disease in Denmark in 2018[53]. Regions of genomic similarity were inferred using BLAST and plotted using Easyfig v2.2.3[79]. The grey gradient indicates the percent identity in the legend. The same prophage was detected in an additional 19 SDSE isolates spanning 10 distinct genome clusters and 7 countries indicating significant dispersion of the prophage in the SDSE population. Source data are provided as a Source Data file.

variation were aligned and assessed using fastGEAR[35]. A total of 526 core genes (45%) had clusters with members from both species consistent with either shared ancestry or whole gene recombination. Putative cross-species recombination was identified in 393 (34%) unique genes, including the MLST genes *gki*, *murI*, and *recP* and the penicillin-binding protein-encoding gene *pbp1b*, which have previously been documented to be affected by inter-species recombination[11,16] (Supplementary Data 2b). Of these, 216 genes had a signature of multiple unique events between species. Recombination affected genes across all functional categories with no significant difference between classes ($p = 0.26$, $\chi^2$ test of independence). While net directionality and absolute frequency of events cannot be inferred using this data, predicted cross-species recombination events affected genes from across the genome with few hot-spot regions of higher density or restriction (Fig. 4b).

Given the genetic similarity between the two organisms, metabolic and virulence regulators are likely also to be shared. Of the standalone regulators, 19/23 (83%) of the regulator genes which are core genes (present in ≥99% of genomes) in *S. pyogenes* were also core in SDSE including *mgc/mgg* which is a homologue to *mga* in *S. pyogenes* (Supplementary Data 1b). Of the two-component regulator genes which are core genes in *S. pyogenes*, 10/12 (83%) were also core in SDSE. While the function and regulatory networks of many of these genes are yet to be defined in SDSE, these findings suggest significant overlap in the regulator repertoire of the two species.

### Predicted conserved metabolic pathways between species identified by pangenome analysis

Examination of non-MGE genes unique to the pangenomes of each species found well-defined KEGG modules predicting metabolic differences between the species. Core to SDSE but absent from the *S. pyogenes* pangenome were modules encoding glycogen biosynthesis (M00854) and threonine biosynthesis (M00018). *S. pyogenes* is known to be auxotrophic for threonine and the absence of threonine and glycogen biosynthesis genes may reflect greater host dependence and/ or adaptation. Unique to *S. pyogenes* were multiple genes encoding V/ A-type ATPases (M00159). While additional differences are likely to exist beyond described KEGG modules, 69% of SDSE and 86% of *S. pyogenes* metabolic genes were shared indicating extensive overlap between the species (Supplementary Fig. 9).

### Conservation of leading *S. pyogenes* vaccine candidates in the global SDSE population

Given the extensive overlap in gene content between SDSE and *S. pyogenes*, we next assessed the carriage of 34 leading *S. pyogenes* candidate vaccine antigens[22] (Supplementary Table 2) in the global SDSE population. Of the 26 candidate vaccine antigens using a full or near-full length gene product, 12 were highly represented in both species (present in >99% of isolates at 70% identity, Fig. 5a). Mean amino acid sequence divergence in SDSE for these 12 candidates from

the *S. pyogenes*-derived reference sequence varied from 80.9–99.9% (Supplementary Fig. 10a). Of the four small peptide vaccine candidates, the multivalent N-terminal M protein and multivalent Tee antigen candidates, which were searched at 100% identity, only J8.0 (C-terminal fragment of M protein) was present in all SDSE isolates. However, none were present in >99% of isolates in both species (Fig. 5b). Potential coverage by five of 11 leading preclinical multicomponent vaccines was >99% in both populations (Fig. 5c, Supplementary Table 3).

Shared antigens may provide cross-species vaccine coverage but conversely, interspecies recombination could yield increased antigenic diversity. Interspecies recombination analysis using fastGEAR was carried out on nine full length antigens which had <25% length variation and were highly present in both species. Five candidates, SpyAD, OppA, Shr, PulA and Fbp54, demonstrated signatures of recombination between a single-species cluster and a genome of the other species (Fig. 5d and Supplementary Fig. 10b). Three candidates, SLO, ADI and TF, had sequence clusters containing both species. Alleles of *srtA* were separated by species between SDSE and *S. pyogenes*. Examination of shared clusters found evidence of SDSE isolates which had acquired a TF allele from *S. pyogenes* (Supplementary Fig. 10b). However, in the setting of limited sequence diversity, recombination could not be inferred for SLO and ADI.

Examination of selection pressures of core genes, including shared core genes which constitute vaccine candidates, may shed light on the evolutionary trajectories and antigenic diversity of the two species. The ratio of non-synonymous to synonymous mutations ($d_N/d_S$) was used to infer positive selection across 665 non-recombinogenic SDSE core genes and 376 non-recombinogenic *S. pyogenes* core genes (Supplementary Data 4). We found 13% (84/665) of SDSE genes compared to 3% (13/376) of *S. pyogenes* genes had a $d_N/d_S > 1$ indicative of positive selection. Although most genes were overall under purifying selection, examination for positive selection at each codon or site within each gene found that 84% (556/665) of SDSE genes and 89% (335/376) of *S. pyogenes* genes had at least one site inferred to be under positive selection. There was no functional enrichment for SDSE genes as defined by COG categories under positive selection ($p = 0.43$, $\chi^2$ test of independence). Analysis of 196 non-recombinogenic core genes shared across species yielded similar results. In SDSE and *S. pyogenes*, 77% (150/196) and 85% (166/196) of shared core genes respectively, were inferred to have at least one site under positive selection including 135 genes which had sites under positive selection in both species. These findings suggest that although most genes were overall under purifying selection, there was evidence of positive selection at the codon level across most SDSE and *S. pyogenes* core genes.

Of the shared vaccine antigens, only *srtA* was inferred to not be affected by intra- or inter-species recombination and demonstrated signatures of purifying selection in both species ($d_N/d_S$: SDSE 0.26, *S. pyogenes* 0.18). However, at the codon level, at least two sites were

**Table 1 | Fully assembled mobile genetic elements (MGE) shared at conserved cross-species insertion regions in the global *Streptococcus dysgalactiae* subsp. *equisimilis* (SDSE) and *S. pyogenes* databases**

| SDSE insertion | *S. pyogenes* insertion | MGE | SDSE isolates (countries) | *S. pyogenes* isolates (countries) | Max nucleotide length and identity | Virulence and AMR cargo genes |
|---|---|---|---|---|---|---|
| SDEG_RSO3340–SDEG_RSO3345 (insertion region 17) | SPY_RSO2725–SPY_RSO2975 | Prophage | 1 genome USA | 2 genomes France | 42 kb 100% identity, 99% coverage | Exotoxin *speC* Streptodornase *spd1* |
| SDEG_RSO3450–SDEG_RSO3465 (insertion region 19) | SPY_RSO3010–SPY_RSO3025 | Prophage | 1 genome USA | 2 genomes UK | 42 kb 89% identity, 98% coverage | Streptodornase *sda1* allele with 95% nucleotide identity to MGAS5005 M1T1 *sda1* allele. |
| SDEG_RSO8710–SDEG_RSO8720 (insertion region 45) | SPY_RSO7175–SPY_RSO7185 | Prophage | 12 genomes USA, Australia, Denmark, Norway, Canada, and Portugal | 104 genomes India, New Zealand, Brazil, Australia, China, Hong Kong, Taiwan, UK, Japan, Trinidad, Canada, Lebanon, USA | 41 kb 99% identity, 99% coverage | Streptodornase *sda1* on the *S. pyogenes* M1T1 prophage, ϕ5005.3. |
| SDEG_RSO9545–SDEG_RSO9550 (insertion region 49) | SPY_RSO1170–SPY_RSO1175 | Nested ICE-like and IS/transposon | 1 genome India | 1 genome Canada | 53 kb 90% identity, 98% coverage | Macrolide AMR *ermB* Aminoglycoside AMR *ant(6)-Ia* and *aph(3')-IIIa* Additional IS/transposon in SDSE genome with chloramphenicol AMR *catA*. |

SDSE insertion sites are mapped to reference genome GGS_124 (NC_012891.1) and *S. pyogenes* insertion sites are mapped to reference genome SF370 (NC_002737.2). SDSE insertion region number refers to naming in Fig. 3a. Antimicrobial resistance, AMR; ICE, integrative conjugative element; IS, insertion sequence.

inferred to be under positive selection in each species suggestive of diversification even in the absence of recombination.

## Discussion

With increasing disease control efforts for *S. pyogenes* including vaccine development[26,27,42] and reports of high disease burden associated with invasive SDSE infection[1–8], an improved understanding of the SDSE genomic population structure and overlap between these closely related pathogens provides a new framework for understanding the evolution and prevention of disease associated with infection by these human pathogens.

The population structure of SDSE was found to have many similarities to that of *S. pyogenes* with multiple evolutionary distinct lineages. Recombination of small genomic fragments was found to be a major driver of diversity of the core genome in SDSE with variation in the accessory genome due to MGE-related genes. These features mirror findings described previously for *S. pyogenes*, suggesting similar evolutionary dynamics at a global population level across these pathogens[22].

At the level of the core genome, more than 75% of core genes were shared between each species consistent with previous findings[43]. Despite the extensive overlap, metabolic differences were found in well characterised KEGG modules such as the presence of threonine and glycogen biosynthesis modules which are core in SDSE but absent in *S. pyogenes*. SDSE has on average, a larger genome than *S. pyogenes* (2.1Mbp vs 1.8Mbp). The threonine and glycogen biosynthesis modules were also present in 8 complete *Streptococcus dysgalactiae* subsp. *dysgalactiae* genomes available on RefSeq (accessed 6 March 2023). While theoretical, it is plausible that these metabolic differences may reflect greater levels of genome reduction in *S. pyogenes*, under the assumption that human isolates of SDSE may have more recently diverged from a multi-host reservoir[44].

We found extensive horizontal gene transfer through homologous recombination across the SDSE and *S. pyogenes* core genome, with over a third of SDSE genes demonstrating evidence of cross-species recombination. These findings extend previous recognition of such interspecies recombination of MLST genes[11,17] and is likely a combination of both ancestral and ongoing genetic transfers. While we do not quantify the frequency of cross-species recombination, the extent of recombination is comparable to that reported by Diop et al. who used measures of homoplasy and sliding window-based sequence identity[45]. This dataset may also provide a foundation for development of future methods controlling for population structure to infer net directionality and frequency of interspecies recombination.

Over 25% of MGE insertion regions and MGE clusters were found to be conserved across SDSE and *S. pyogenes*, suggesting much greater sharing of MGEs across the species than previously appreciated. Within MGE clusters, near-identical complete MGEs were detected including ϕ5005.3, the prophage carrying streptodornase Sda1 found in the globally successful M1T1 *S. pyogenes* lineage[41,46]. The prophage was present across multiple distinct lineages in SDSE indicating sharing and dissemination of the element in the SDSE population.

Extending these findings to *S. pyogenes* vaccine candidates, 12 of 34 antigens and five of 11 multicomponent vaccine candidates were predicted to contain antigens present in >99% of isolates from both species. Of the 12 antigens, 6 had evidence of interspecies recombination including components of multivalent vaccines, SpyAD, OppA, TF and PulA. This suggests that SDSE may represent an additional reservoir for antigenic diversity particularly if vaccine candidates enhance selection in an antigenically diverse region. Even greater diversity likely exists considering current limited whole genome sampling of SDSE in the continents of Africa and South America. Thus, surveillance of SDSE should be considered in the context of *S. pyogenes* vaccine development and monitoring.

Although published data on preclinical efficacy of vaccine candidates in SDSE is limited beyond preliminary data on the J8

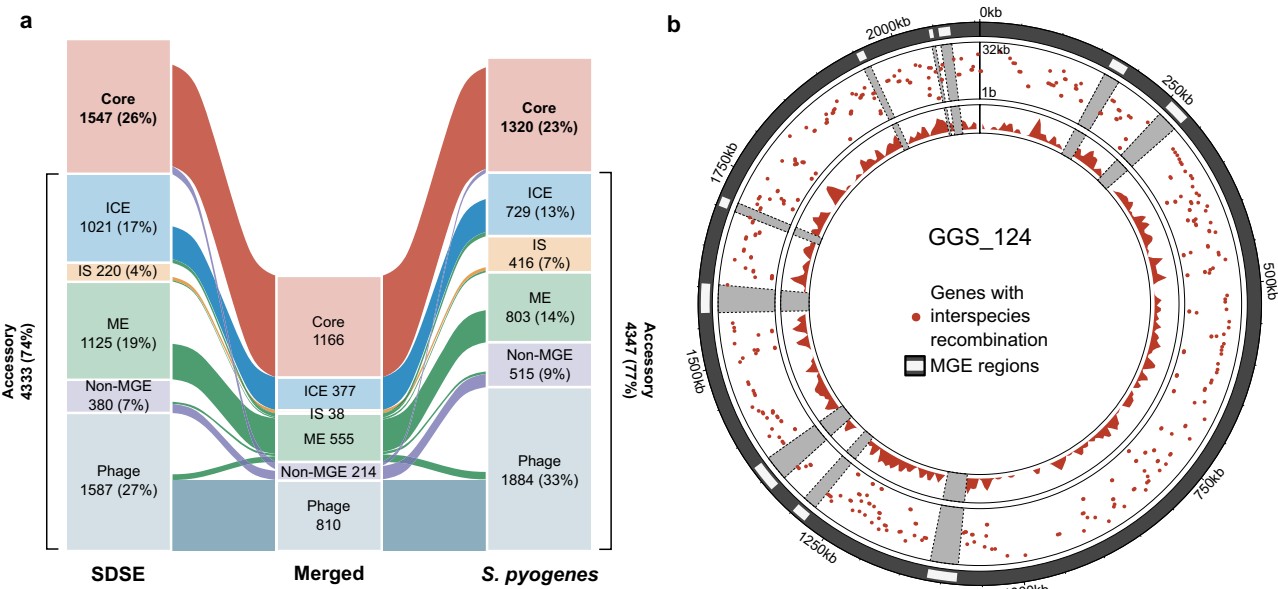

**Fig. 4 | Comparison and recombination signatures between the *Streptococcus dysgalactiae* subsp. *equisimilis* (SDSE) and *S. pyogenes* pangenome. a** Alluvial plot of the shared SDSE (*n* = 501) and *S. pyogenes* (*n* = 2083) pangenome. Categorisation of core genes (present in ≥99% of isolates in both species) and accessory genes by association with mobile genetic element (MGE). ICE, integrative conjugative element; IS, insertion sequence; ME, mobility element. Genes classified as belonging to different types of MGEs or an MGE/non-MGE combination across the species were binned as ME in the merged pangenome. Unclassified genes were excluded and a small number of genes were merged when overlaying pangenomes resulting in a slightly smaller merged pangenome than the pangenomes of

individual species. **b** Circular rainfall plot of core genes with signatures of recombination flagged by fastGEAR[35] plotted relative to the GGS_124 reference genome (NC_012891.1). Genes flagged with putative interspecies recombination events are highlighted by red points. The distance between each gene with its neighbour within the same category is plotted on a $\log_{10}$ scale between 1 bp to 32 kbp. The innermost track plots the density of genes with evidence of interspecies recombination using a window size of 10,000 bp. MGE regions are masked in grey. The rainfall plot demonstrates that genes flagged as affected by inter-species recombinants are dispersed across the SDSE genome. Source data are provided as a Source Data file.

peptide[47], the prevalence of conserved antigens suggest there may be cross-species effects of vaccines intended to target *S. pyogenes*. However, the sequence-based approach used in this study does not consider potential conserved structural epitopes which may confer antibody cross-reactivity or cross-opsonisation to divergent alleles. It should be noted though that an immunological correlate of protection has yet to be determined for *S. pyogenes* or SDSE. Additionally, this study infers potential shared cross-species vaccine antigens based on sequence homology which does not consider other factors which may affect cross-species vaccine effectiveness including levels of expression of these antigens or potential immunogenicity differences across species. However, the striking similarities of many of these candidates across species warrants further investigation of their cross-species effects.

This systematic and detailed analysis of the overlap between SDSE and *S. pyogenes* at the level of the core genome, recombination, and MGEs reveals the extensive shared genomic content between these closely related pathogens and provides a platform for further investigations into their shared biology. Genomic exchange however is not limited to movement between these two organisms. Indeed, horizontal gene transfer between SDSE and another beta-haemolytic *Streptococcus*, *Streptococcus agalactiae*, has been described and elements of SDSE biology such as a second FCT locus, are more closely associated with *S. agalactiae* than *S. pyogenes*[34]. These methods could therefore be applied to other closely related species to provide insight into the shared biology between disease-causing streptococci in humans.

## Methods
### Bacterial isolates
The collection of 501 global SDSE sequences included publicly available short-read sequence data from NCBI sequence read archive (SRA)

and complete genome assemblies from NCBI RefSeq until 4 May 2022. These included studies of invasive and non-invasive SDSE from Japan[43,48,49], Germany[50], India[51], China[52], Canada[23], Norway[24,34,44], USA[18,19], Denmark[53], Switzerland[54] and The Gambia[55]. A further 228 invasive and non-invasive isolates were collected from Australia, France, USA, India, Argentina, Trinidad, Japan, Fiji, and Portugal to maximise geographical diversity. Isolates were obtained from retrospective isolate collections[11,56], surveillance studies[8,57,58], and routine clinical care. Metadata presented in this study were anonymised. Additional institutional review board approval was not required for sequencing of bacterial strains with anonymised data. Metadata for the isolates are available in Supplementary Data 1.

### Genome sequencing, assembly and quality control
For the newly sequenced genomes, microbial DNA was extracted and 75–100 bp paired-end libraries were sequenced using the Illumina HiSeq 2500 platform (The Wellcome Sanger Institute, United Kingdom). Reads were trimmed using Trim Galore v0.6.6 (https://github.com/FelixKrueger/TrimGalore) with a Phred score threshold of 20–25 and filtered to remove reads <36 bp. Reads were examined for contamination using Kraken2 v2.1.2[59]. Draft assemblies were generated using Shovill v1.1.0 (https://github.com/tseemann/shovill) with SPAdes assembler v3.14.0[60] and a minimum contig length of 200 bp. Only assemblies with <150 contigs (mean 94, range 57–149), total assembly size between 1.9–2.4 Mb (mean, 2.11 Mb, range 1.91–2.30 Mb), and GC% between 38–40% (mean 39.3%, range 38.7–39.6%) were included. The mean N50 was 72,091 bp (range 42,987–146,734 bp). Annotations were generated using Prokka v1.14.6[61] using the '–proteins' flag with four Refseq *S. pyogenes* and SDSE genomes to supplement annotations for consistency. *emm* sequence typing was performed using emmtyper v0.2.0 (https://github.com/MDU-PHL/emmtyper) and MLST assigned using MLST v2.22.0 (https://github.com/tseemann/mlst)[62]. The

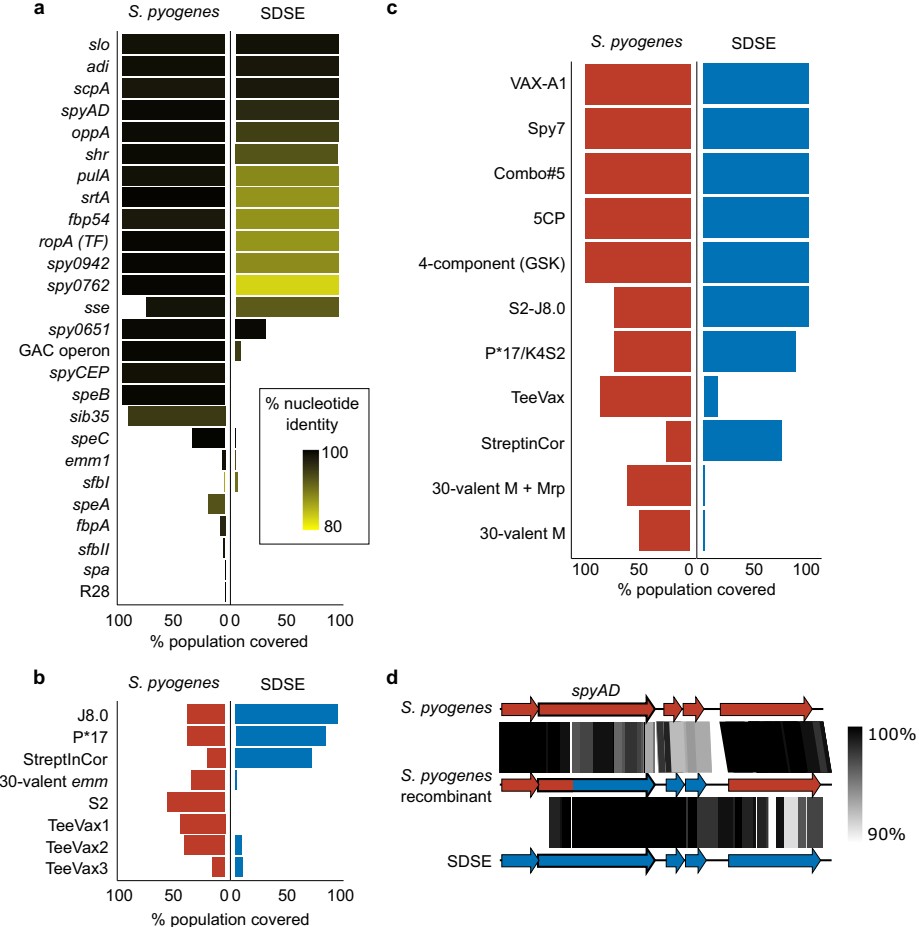

**Fig. 5 | Theoretical coverage of preclinical vaccine candidates and multicomponent formulations in global _Streptococcus dysgalactiae_ subsp. _equisimilis_ (SDSE, _n_ = 501) and _S. pyogenes_ (_n_ = 2083) populations. a** Whole gene candidates were screened in both species at 70% nucleotide length and identity. Theoretical coverage of the population (% presence) is expressed by length of bar and conservation relative to the _S. pyogenes_ query sequence is reflected by gradient fill from 80% similar (yellow) to 100% (black). **b** Peptides and gene sub-domain candidates were screened without calculation of sequence diversity. Peptide fragments were screened using a 100% match approach and a six-frame translation of the sequence (J8.0, S2, P*17, StreptInCor 'common epitope') and gene sub-domains (30-valent _emm_, T antigen fragments in TeeVax1, TeeVax2, TeeVax3) were screened

at 70% nucleotide identity and coverage. **c** Theoretical coverage of SDSE and _S. pyogenes_ isolates in this study by multivalent vaccine candidates. **d** Exemplar of putative recombination between _S. pyogenes_ and SDSE vaccine candidate _spyAD_. _S. pyogenes_ isolate NS1140 demonstrated evidence of cross-species recombination around the vaccine candidate _spyAD_ gene. Sequence similarity between _S. pyogenes_ reference genome SF370 NC_002737.2 (top), recombinant _S. pyogenes_ isolate NS1140 (middle), and SDSE reference 3836-05 (bottom) demonstrated greater similarity between NS1140 and 3836-05 at position 419 of _spyAD_ to 2 ORFs downstream. Regions of genomic similarity were inferred using BLAST and plotted using Easyfig v2.2.3[79]. Source data are provided as a Source Data file.

Lancefield group carbohydrate was inferred by the presence genes in the 14 gene group C carbohydrate synthesis locus[18], 15 gene group G[15] locus and 12 gene group A[15] locus in the SDSE pangenome. The carbohydrate synthesis locus was present in a conserved location between core genes _mscF_ (SDEG_RS03715) and _pepT_ (SDEG_RS03795). Reads for all newly sequenced genomes are available on SRA with accession numbers provided in Supplementary Data 1.

To make a complete genome sequence of the _S. dysgalactiae_ subsp. _equisimilis_ strain NS3396, high quality genomic DNA was extracted using the GenElute Bacterial Genomic DNA Kit (Sigma). The complete genome assembly of NS3396 (GenBank accession CP128987) was performed using SMRT analysis system v2.3.0.140936 (Pacific Biosciences). Raw sequence data was de novo assembled with the HGAP3 protocol. Polished contigs were error corrected with Quiver v1, and the final assembly structure checked by mapping raw reads against the alignment with BridgeMapper v1 as previously described[63].

Complete genomes were checked for genome arrangement and duplications around RNA operons using socru v2.2.4[64]. Genomes with unverified large-scale rearrangements and duplications around RNA

operons compared to published complete genomes NC_012891.1 GGS_124, NC_018712.1 RE378 and NC_019042.1 AC-2713 were excluded. Four complete genomes were included in the final database.

**Phylogenetic analysis**

Recombination masked distances between SDSE genomes were calculated using Verticall v0.4.0 (https://github.com/rrwick/Verticall) which conducts pairwise comparisons between genomes and masks regions with increased and/or decreased sequence divergence as putatively recombinogenic. Using recombination masked distances, a minimum evolution phylogenetic tree was generated using fastME v2.1.6.1[65] with nearest neighbour interchange using BME criterion for optimisation and subtree pruning and regrafting.

A comparative maximum-likelihood tree, without masking of recombination, was generated by alignment of all genomes against reference genome NC_012891.1 GGS_124 using Snippy v4.6.0 (https://github.com/tseemann/snippy). MGE regions were masked and tree inference conducted with IQ-tree v2.0.6[66] using a GTR + F + G4 model. Tree comparisons were performed using TreeDist v2.5.0[67] to calculate

generalised Robinson-Foulds distances with mutual clustering information and matched splits were visualised with the 'VisualizeMatching' function.

Bayesian dating of the MRCA of the two most sampled genome clusters (groups '1' and '2') was performed using BactDating v1.1[30] with an additive relaxed clock model, coalescent prior, and $10^8$ iterations to ensure Markov Chain Monte Carlo convergence with parameter effective sample size >190 after 50% burn-in. Recombination-masked alignments were used as input. Alignments were generated using Snippy v4.6.0 against a reference genome within the same genome cluster when available, NC_018712.1 for genome cluster '2', or a high-quality draft genome (SRR3676046) for genome cluster '1'. Regions affected by recombination were masked using Gubbins v3.1.2[31] and IQ-TREE with a GTR + G4 model for tree building, and set with maximum 10 iterations, minimum of 5 SNPs to identify a recombination block, minimum window size of 100 bp and maximum window size of 10,000 bp. Contigs were padded with 10,000 Ns, the maximum window size, for the genome cluster '1' alignment to prevent Gubbins calling recombination across contigs. Four non-stG62647 isolates were excluded from genome cluster '1' for dating analysis as they were >1500 SNPs distant and formed a distinct sub-lineage within the cluster.

## SDSE population genomics
Evolutionarily related clusters (genome clusters) were defined using PopPUNK v2.4.0[29] which has previously been used to describe the population genomics of *S. pyogenes*. PopPUNK assigns clusters based on core and accessory distances calculated using sliding k-mers. Distances were calculated using k-mer sizes between 13 and 29 at steps of four (Supplementary Fig. 2). A three-distribution Bayesian Gaussian Mixture Model (BGMM) was fit with 2D cluster boundary refinement to obtain a network score of 0.93 and was chosen after benchmarking against a model built using HDBSCAN and BGMM models with different numbers of distributions (Supplementary Fig. 2b, c). The PopPUNK model (v1) and genome cluster designations from this study are available at https://www.bacpop.org/poppunk/ and can be iteratively expanded.

Genomic distances used to compare PopPUNK genome clusters with traditional epidemiological markers, *emm* and MLST, were recombination masked distances calculated by Verticall as above. Distances are analogous to 100% minus average nucleotide identity. Single locus variant MLST clonal complexes were calculated using the goeBURST algorithm as implemented in PHYLOViZ v2.0[68].

Known SDSE and *S. pyogenes* virulence factors were screened in the SDSE database using screen_assembly v1.2.7[22] with sequence accessions as listed in Supplementary Data 1b at 70% nucleotide identity and 70% coverage. Virulence factors were supplemented using Abricate v1.0.1 (https://github.com/tseemann/abricate) with VFDB[69] at 70% nucleotide identity and 70% coverage. Genes with length variation and assembly breaks due to large repeat regions, were screened using Magphi v1.0.1[70] for conserved 5' and 3' sequences at a distance equal to the maximum known length of the gene. Antimicrobial resistance genes were screened using Abritamr v1.0.6 (https://github.com/MDU-PHL/abritamr), a wrapper for AMRfinder plus v3.10.18[71], at default 90% identity and 50% coverage.

## SDSE pangenome and MGE analysis
The SDSE pangenome was defined using Panaroo v1.2.10[72] which utilises a pangenome graph-based approach for pangenome clustering. Panaroo was run in 'strict' mode with initial clustering at 98% length and sequence identity followed by a family threshold of 70% to collapse syntenic gene families. Core genes were defined as genes present in ≥99% of genomes, shell accessory genes between 15%-99% of genomes, and cloud accessory genes in <15% of genomes. COG functional categories were assigned to core genes using eggNOG-mapper v2.1.7[73]

with default Diamond mode. COG categories J, K, L, A, B and Y were summarised as genes involved in 'information storage and processing', categories T, D, V, U, M, N, O, W and Z were summarised as genes involved in 'cellular processing and signalling', categories C, G, E, F, H, I, P and Q were summarised as genes involved in 'metabolism'.

MGE detection and classification was performed by adapting an automated classification algorithm by Khedkar et al.[36] and enhanced by Corekaburra v0.0.2[74]. Corekaburra maps core gene consensus synteny from pangenomes and was used to find stretches of accessory genes or 'accessory segments' between core genes which are used as anchor points. Full details of the logic for classification of MGEs is as described by Khedkar et al.[36] and Hidden Markov Models (HMMs) are available at http://promge.embl.de/. Briefly, accessory segments were investigated for integrase/recombinase subfamilies using HMMER v3.3.2[75] 'hmmsearch' with model-specific gathering thresholds. Representative translated protein sequences from the pangenome were used and resulted in almost identical results compared to searches using sequences from individual genomes. Integrase/recombinase subfamilies were mapped to specific MGE classes: prophage, ICE, IS/transposon and ME (or mobility islands). For subfamilies associated with more than one MGE class, additional information including presence of prophage or ICE structural genes was required. Prophage structural genes were annotated using eggNOG-mapper v2.1.7[73]. At least two prophage genes with an appropriate integrase/recombinase were required to classify an element as a prophage. ICE structural genes were classified using HMMs from TXSScan[76] and 'hmmsearch' with an E-value threshold of 0.001. Unlike the original method by Khedkar et al., we required the presence of at least one coupling protein and one T4SS ATPase with an appropriate integrase/recombinase for classification of an ICE to improve specificity. Elements with a prophage specific integrase/recombinase, but no prophage structural genes were classified as phage-like. Accessory segments without enough contextual information or which contained both prophage and ICE structural genes were classified as ME. ME may also represent degraded elements, integrative mobilizable elements (IME), or accessory segments fragmented by sequence breaks with a disconnect between the integrase/recombinase and structural genes. The boundaries of nested segments with more than one recombinase were not resolved and we did not define the exact *attR* and *attL* sites.

Classification of accessory genes into non-MGE or MGE classes was based on the frequency each gene was detected in each class. Accessory segments with prophage or ICE structural genes but no integrase or unclassified segments with a sequence break did not contribute to the count as they may represent part of a fragmented MGE. Genes were classified as MGE-related if it appeared on any MGE more frequently than it was found on a non-MGE element. A small number of genes only present adjacent to contig breaks without an accompanying integrase were considered unclassified as they could not be confidently grouped into an MGE or non-MGE category. For the purposes of gene classification, phage-like elements were grouped with prophage.

## SDSE recombination detection
To examine evidence of recombination within the core SDSE genome, fastGEAR[35] was run on 1543 core gene alignments from the 501 SDSE strains included in the study. Gene alignments were performed using MAFFT v7.505[77]. Alignments with greater than 25% gap characters were excluded. fastGEAR infers population structure for each alignment, allowing for the detection of lineages or clusters that have 'ancestral' and 'recent' recombination events between them. Default parameters were used with a minimum threshold of 4 bp applied for the recombination length.

The relative ratio of mutation due to recombination to vertically inherited mutation (r/m) was determined for the 12 most frequently sampled genome clusters (385 isolates) using Gubbins v3.1.2[31] as

described above for phylogenetic and BactDating analysis. Complete genomes within the same genome cluster were available and used for alignment for four genome clusters. The remaining genome clusters were aligned against a high-quality draft genome within the same genome cluster. MGE regions were masked in the alignment. The r/m, number of recombination events and length of recombination segments was calculated for each genome within a genome cluster by summing along branches from the Gubbins output. The median r/m was calculated for each genome cluster, and the median of these values was given as the species r/m.

### MGE insertion site mapping across SDSE and *S. pyogenes*

SDSE MGE insertion regions were defined between two core genes using Corekaburra[74] to call pangenome synteny. Only prophage, ICE and ME insertion sites were mapped. MGE counts presented in Supplementary Data 3a were estimated by summarising fragmented MGEs on the same contig as a core gene, to the corresponding core gene pair insertion region. Phage-like elements were grouped with prophages. Alternative insertion sites were defined as the less commonly found connection between two core genes and represent genome rearrangements around MGEs.

Insertion regions found in *S. pyogenes* were collated from McShan et al.[37] and Berbel et al.[38] in addition to 13 newly reported insertion sites using Corekaburra. Insertion regions were mapped across species by merging the SDSE and *S. pyogenes* pangenome. The *S. pyogenes* pangenome was defined using Panaroo v1.2.10[72] with the same parameters as that used for SDSE. The pangenomes of the two species were then merged using 'panaroo-merge' which overlays pangenome graphs. Merging was performed with an initial clustering threshold of 90% identity and 90% length followed by a threshold of 70% for collapsing syntenic genes. Insertion sites in each species were then matched using the merged pangenome. Where no match was obtained, including where genes were core in only one species, core genes within two genes either side of the insertion region were manually inspected for a match.

### Shared genes and MGEs at insertion sites

MGE genes at shared insertion sites were determined using Corekaburra[74] by listing all accessory genes present between core gene insertion sites which were conserved across SDSE and *S. pyogenes*. Genes were considered shared across SDSE and *S. pyogenes* if they overlapped in the merged pangenome.

MGEs were clustered using mge-cluster v1.0.2[39] which maps distances between MGEs using presence and absence of shared unitigs, projects the distances in two dimensions using t-SNE and clusters similar elements using HDBSCAN. mge-cluster has been used with plasmid sequences and we now extend it to use with prophage and ICE. mge-cluster was run with t-SNE perplexity of 75 and HDBSCAN minimum cluster size of 30. MGE sequences were extracted using Magphi v1.0.1[70] with insertion site core genes used as seed sequences, maximum seed distance determined by the largest distance between the respective core pair calculated by Corekaburra, and '--print_breaks' selected which attempts to extract sequences across contig breaks. When cross-species seed hits could not be obtained using nucleotide sequences, input of the '--protein_seed' flag with translated protein sequences were used. To ensure only high quality MGE sequences were obtained, only sequences longer than 10–15 kb were included. Fragmented MGEs were concatenated for input into mge-cluster. Some segments were unable to be extracted by Magphi as seed core genes were present on small contigs resulting in difficulty determining whether to extract sequences in the 5' or 3' direction.

Highly similar MGEs present in SDSE and *S. pyogenes* were found by sequence-based clustering using CD-HIT v4.8.1[78] 'cd-hit-est' with word size 5, sequence identity threshold 0.8 and length difference cut-off 0.8. MGE alignment figures were generated using Easyfig v2.2.3[79].

### Comparing SDSE and *S. pyogenes* pangenome

The accessory gene classification scheme described for SDSE and COG annotations were mapped to the *S. pyogenes* pangenome. The merged pangenome was then used to map functional classes of core genes and accessory gene MGE classes across species. Panaroo[72] combines a small number of genes/coding sequences when merging pangenomes resulting in slightly fewer pangenome genes in the merged pangenome compared to that for the individual species.

Metabolic differences between SDSE and *S. pyogenes* were inferred by searching for well-defined complete KEGG modules present in one species but not the other. Non-MGE and core genes unique to each species were extracted from the pangenomes of each respective species and KO identifiers were assigned using eggNOG-mapper v2.1.7[73]. KO numbers of unique genes from both species were then mapped simultaneously to KEGG modules using KEGG mapper Reconstruct[80].

### Recombination detection across species

Gene alignments of the 1116 shared core genes of SDSE and *S. pyogenes* with <25% gap characters were created and analysed with fastGEAR[35]. Interspecies recombination was inferred by two criteria from the fastGEAR results. Lineages or sequence clusters inferred by fastGEAR could contain sequences from one species or both. A predicted recombination event from a cluster containing only one species to a genome from the other species, an event classified as 'recent' by fastGEAR, was classified as a putatively cross-species. The presence of both species within the same lineage or cluster when more than one lineage was predicted, could occur in the setting of cross-species recombination of the whole gene or with shared ancestry. As these could not be easily separated, cross-species clusters were recorded separately, and locus tags are provided in Supplementary Data 2b for individual interrogation.

### Vaccine antigen screen

*S. pyogenes* vaccine candidates were screened for presence and sequence diversity in SDSE. Vaccine candidates and screening methods were adapted from a previous report of vaccine antigenic diversity in *S. pyogenes* and updated with new multivalent antigens and formulations[22,42] (Supplementary Table 2). The presence of vaccine antigen genes was determined using screen_assembly v1.2.7[22] with a cut-off of 70% nucleotide identity and 70% coverage. Sequence diversity was presented as nucleotide divergence calculated by BLASTn or predicted amino acid divergence using tBLASTn as indicated.

For the 30-valent M protein vaccine, the 180 bp hypervariable 5' sequence was extracted from publicly available databases and compared against the N-terminal sequence of SDSE *emm* types represented in the 501 genomes from this report at 70% nucleotide identity and 70% coverage. The representative/type SDSE *emm* sequence (e.g., *stG*62647.0, *stG*840.0) was used for the comparison. For the T antigens, nucleotide sequences of the individual subdomains from different T alleles included in the fusion multivalent vaccine formulations were extracted and searched at a threshold of 70% nucleotide identity and 70% coverage. Results were presented as presence/absence as M and T antigens are hypervariable.

The small peptide antigens J8.0, StrepInCor 'common' overlapping B and T cell epitope, P*17 and S2 were screened using a six-frame translation of the target genome and search at 100% identity and coverage. As P*17 had two amino acid substitutions at positions 13 and 17 introduced which are not naturally occurring, amino acids at these positions were replaced with a wildcard for the search.

### Signatures of molecular adaptation

Molecular signatures of adaptation were inferred by the ratio of non-synonymous ($d_N$) to synonymous ($d_S$) mutations, $d_N/d_S$ or $\omega$, using a codon model implemented in HyPhy v2.5.58[81]. Only non-recombinogenic core genes as inferred by fastGEAR[35] in this study

(SDSE, $n = 665$) or previously for *S. pyogenes*[22] ($n = 376$) were included after exclusion of highly fragmented genes as indicated by sequences with ≥25% gaps which could affect the fit of the codon model. Of the core genes analysed, 196 were core in both species. A global $d_N/d_S$ was estimated across the entire alignment for each gene which allows for a straightforward comparison between genes and species. As most genes are expected to be under purifying selection ($d_N/d_S < 1$) when averaged across their entire alignment, evidence of pervasive (across the entire phylogeny) positive selection ($d_N/d_S > 1$) was investigated for each site (codon) for each gene using FUBAR[82] as implemented in HyPhy. The species phylogeny was used for each model and sites with a posterior probability ≥0.9 were counted as putatively under positive selection in the site-specific analysis. Due to sampling differences between the SDSE and *S. pyogenes* datasets, the selection analyses should be interpreted carefully as sites inferred to be under selection may be influenced by the sampling and diversity included.

### Reporting summary

Further information on research design is available in the Nature Portfolio Reporting Summary linked to this article.

## Data availability

Reads for newly sequenced genomes have been deposited in the European Nucleotide Archive (https://www.ebi.ac.uk/ena/). Accessions for raw sequencing data are available in Supplementary Data 1a. The complete genome sequence of *S. dysgalactiae* subsp. *equisimilis* NS3396 was deposited to GenBank (accession CP128987, https://www.ncbi.nlm.nih.gov/genbank/). Source data are provided with this paper. The authors confirm all supporting data have been provided within the article or in supplementary data files. Source data are provided with this paper.

## Code availability

Supplementary code used to extract and classify accession segments and MGEs is available at https://github.com/OuliXie/Global_SDSE. (https://doi.org/10.5281/zenodo.10669012).

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

## Acknowledgements

The authors would like to acknowledge that a portion of samples were collected from Aboriginal Australian populations which are severely impacted by streptococcal associated disease, and we hope that this study can add to the evidence base assisting in the prevention of such diseases. The work was supported by the National Health and Medical Research Council of Australia (NHMRC) and The Wellcome Trust, UK. MRD was supported by a NHMRC postdoctoral training fellowship (635250) and a University of Melbourne CR Roper Fellowship. OX was supported by the NHMRC postgraduate scholarship (GNT2013831) and Avant Foundation Doctors in Training Research Scholarship (2021/000017). We acknowledge the assistance of the sequencing and pathogen informatics core teams at the Wellcome Sanger Institute, UK where this work was supported by the Wellcome Trust core grants 206194 and 108413/A/15/D.

## Author contributions

O.X., D.J.M. and M.R.D. planned the study. R.J.T., L.S., K.S.S., T.H., P.G., A.C.S., M.R.B., B.W.B., M.D.P., M.R., D.E.B., B.J.C. and D.J.M. provided samples and metadata. O.X., J.M.M., A.J.H., M.G.J., J.A. Le., N.L.B.Z., O.B., S.L.B., G.P.C., G.T.H., L.M., J.A.La., T.B.J., S.A.B., G.D., S.D.B., M.J.W., S.Y.C.T., D.J.M. and M.R.D. designed experimental procedures and generated data. O.X., J.M.M., A.J.H., M.G.J., J.A.Le., N.L.B.Z., S.L.B., G.P.C., S.Y.C.T., D.J.M. and M.R.D. analysed data. O.X., J.M.M., A.J.H., S.L.B., G.P.C., D.J.M. and M.R.D. wrote the manuscript. All authors revised and approved the manuscript.

## Competing interests

The authors declare no competing interests.

## Additional information

[1]Department of Infectious Diseases, The University of Melbourne at the Peter Doherty Institute for Infection and Immunity, Melbourne, Australia. [2]Monash Infectious Diseases, Monash Health, Melbourne, Australia. [3]Department of Microbiology and Immunology, The University of Melbourne at the Peter Doherty Institute for Infection and Immunity, Melbourne, Australia. [4]Menzies School of Health Research, Charles Darwin University, Darwin, Australia. [5]European Molecular Biology Laboratory, European Bioinformatics Institute EMBL-EBI, Hinxton, Cambridgeshire, UK. [6]Australian Infectious Diseases Research Centre and School of Chemistry and Molecular Biosciences, The University of Queensland, Brisbane, Australia. [7]Doherty Applied Microbial Genomics, Department of Microbiology and Immunology, The University of Melbourne at the Peter Doherty Institute for Infection and Immunity, Melbourne, Australia. [8]Department of Biostatistics, University of Oslo, Oslo, Norway. [9]Faculty of Veterinary Science, The University of Sydney, Sydney, Australia. [10]Infection and Inflammation Program, QIMR Berghofer Medical Research Institute, Brisbane, Australia. [11]School of Science & Technology, University of New England, Armidale, Australia. [12]Department of Bacteriology, Nagoya City University Graduate School of Medical Sciences, Nagoya, Japan. [13]Tropical Diseases, Murdoch Children's Research Institute, Parkville, Australia. [14]Institute for Glycomics, Griffith University, Southport, Australia. [15]Respiratory Disease Branch, National Center for Immunizations and Respiratory Diseases, Centers for Disease Control and Prevention, Atlanta, GA, USA. [16]Instituto de Microbiologia, Instituto de Medicina Molecular, Faculdade de Medicina, Universidade de Lisboa, Lisboa, Portugal. [17]Department of Pathology, Microbiology and Immunology, New York Medical College, Valhalla, NY, USA. [18]Parasites and Microbes, Wellcome Sanger Institute, Hinxton, Cambridgeshire, UK. [19]Institute for Molecular Bioscience, The University of Queensland, Brisbane, Australia. [20]Victorian Infectious Disease Service, The Royal Melbourne Hospital at the Peter Doherty Institute for Infection and Immunity, Melbourne, Australia. [21]School of Science, Technology and Engineering, and Centre for Bioinnovation, University of the Sunshine Coast, Sippy Downs, Australia. [22]These authors contributed equally: David J. McMillan, Mark R. Davies. ✉e-mail: mark.davies1@unimelb.edu.au

