## [Peer Review File · Nature Communications]

Inter-species gene flow drives ongoing evolution of
Streptococcus pyogenes and *Streptococcus dysgalactiae*
subsp. *equisimilis*Editorial Note: This manuscript has been previously reviewed at another journal that is not operating a transparent peer review scheme. This document only contains reviewer comments and rebuttal letters for versions considered at *Nature Communications*.

REVIEWERS' COMMENTS

Reviewer #1 (Remarks to the Author):

The authors have addressed the main points raised previously. There are some minor things to highlight

Lines 302-305 "19/23 (83%) of the regulator genes which are core genes (present in $\geq 99\%$ of genomes) in *S. pyogenes* were also core in SDSE including *mgc/mgg* which is a homologue to *mga* in *S. pyogenes*. Of the two-component regulator genes which are core genes in *S. pyogenes*, 9/11 (82%) were also core in SDSE". I could not see these data in a Table- can the authors supply this?

In the Discussion (line 378-) in relation to the proportion of core genes that are shared, the authors should perhaps acknowledge the other literature that has reported this same/similar figure?

The authors report on the presence of genes encoding a number of potential vaccine candidates, however none of these have been verified as expressed by SDSE (at least not in this report). This ought to be commented on as a limitation of a purely genomic study.

Reviewer #4 (Remarks to the Author):

Xie et al have revised the manuscript and I appreciate the implementation of the dN/dS analysis. While an overall dN/dS < 1 (indicating purifying selection) is what I'd expect, the large amount of codons with evidence for positive selection is surprising. Unfortunately no enrichment in functional gene groups was detected. I was wondering how the authors inferred gene groups, but could not find it in the Methods section. Maybe I missed it, but in case not, I'd suggest that to be added. Overall I think the revised manuscript is a comprehensive genomic comparison of two closely related species *Streptococcus dysgalactiae* and *Streptococcus pyogenes*, relevant for the scientific infectious biology and *Streptococcus* community.

Response to reviewer comments

Inter-species gene flow drives ongoing evolution of *Streptococcus pyogenes* and *Streptococcus dysgalactiae* subsp. *equisimilis*

Reviewer 1

1. Lines 302-305: 19/23 (83%) of the regulator genes which are core genes (present in >99% of genomes) in *S. pyogenes* were also core in SDSE including *mgc/mgg* which is a homologue to *mga* in *S. pyogenes*. Of the two-component regulator genes which are core genes in *S. pyogenes*, 9/11 (82%) were also core in SDSE... I could not see these data in a Table- can the authors supply this?

Apologies the data is available in Supplementary Table 1b which is difficult to navigate in pdf form. We have added a reference to Supplementary Data 1b. The original data table is also submitted as an Excel spreadsheet which should be available with the publication version of the article.

We have also noticed an error in the exact numbers provided – we have amended “9/11 (82%)” to “10/12 (83%)”.

2. In the Discussion (line 378-) in relation to the proportion of core genes that are shared, the authors should perhaps acknowledge the other literature that has reported this same/similar figure?

We have added “consistent with previous findings” to line 378 and have cited the paper by Shimomura et al.

3. The authors report on the presence of genes encoding a number of potential vaccine candidates, however none of these have been verified as expressed by SDSE (at least

not in this report). This ought to be commented on as a limitation of a purely genomic study.

Thank you for this comment. Many of these antigens have been shown in previous studies to be expressed in SDSE e.g., SLO, scpA. We have added a sentence regarding expression of these genes in lines 423-424.

Reviewer 4

1. Xie et al have revised the manuscript and I appreciate the implementation of the dN/dS analysis. While an overall dN/dS < 1 (indicating purifying selection) is what I'd expect, the large amount of codons with evidence for positive selection is surprising. Unfortunately no enrichment in functional gene groups was detected. I was wondering how the authors inferred gene groups, but could not find it in the Methods section. Maybe I missed it, but in case not, I'd suggest that to be added.

Thank you for pointing this out. Functional categories were inferred using COG categories as described in the pangenome section of the methods. We have added a clarification that the functional categories referred to in the dN/dS section are COG categories.